# Lipid Profile, Lipoprotein Subfractions, and Fluidity of Membranes in Children and Adolescents with Depressive Disorder: Effect of Omega-3 Fatty Acids in a Double-Blind Randomized Controlled Study

**DOI:** 10.3390/biom10101427

**Published:** 2020-10-08

**Authors:** Barbora Katrenčíková, Magdaléna Vaváková, Iveta Waczulíková, Stanislav Oravec, Iveta Garaiova, Zuzana Nagyová, Nataša Hlaváčová, Zdenka Ďuračková, Jana Trebatická

**Affiliations:** 1Institute of Medical Chemistry, Biochemistry and Clinical Biochemistry, Faculty of Medicine, Comenius University, Sasinkova 2, 813 72 Bratislava, Slovakia; katrencikov2@uniba.sk (B.K.); magdalena.vavakova@med.lu.se (M.V.); 2Department of Nuclear Physics and Biophysics, Faculty of Mathematics, Physics and Informatics, Comenius University, Mlynská dolina F1, 842 48 Bratislava, Slovakia; waczulikova@gmail.com; 31st Department of Internal Medicine, Faculty of Medicine, Comenius University, Mickiewiczova 13, 81369 Bratislava, Slovakia; stanislavoravec@yahoo.com; 4Research and Development. Cultech Ltd., Unit 2 Christchurch Road, Port Talbot SA12 7BZ, UK; ivetag@cultech.co.uk; 5Pediatric Center, Juvenalia, s.r.o, Veľkoblahovská 44A, 929 01 Dunajská Streda, Slovakia; juvenaliads@gmail.com; 6Institute of Experimental Endocrinology, Biomedical Research Center of the Slovak Academy of Sciences, Dúbravská cesta 9, 945 05 Bratislava, Slovakia; natasa.hlavacova@savba.sk; 7Department of Child and Adolescent Psychiatry, Faculty of Medicine and The National Institute of Children’s Diseases, Comenius University, Limbová 1, 833 40 Bratislava, Slovakia; jana.trebaticka@fmed.uniba.sk

**Keywords:** depressive disorder, omega-3 fatty acids, lipid profile, fluidity of membrane, children and adolescents

## Abstract

Depressive disorder (DD) is a psychiatric disorder whose molecular basis is not fully understood. It is assumed that reduced consumption of fish and omega-3 fatty acids (FA) is associated with DD. Other lipids such as total cholesterol (TCH), LDL-, and HDL-cholesterols (LDL-CH, HDL-CH) also play a role in depression. The primary endpoint of the study was the effect of omega-3 FA on the severity of depression in children and adolescents. This study aimed to investigate the secondary endpoint, relationship between depressive disorder symptoms and lipid profile, LDL- and HDL-cholesterol subfractions, Paraoxonase 1 (PON1) activities, and erythrocyte membrane fluidity in 58 depressed children and adolescents (calculated by the statistical program on the effect size), as well as the effect of omega-3 FA on the monitored parameters. Depressive symptoms were assessed by the Children’s Depression Inventory (CDI), lipid profile by standard biochemical procedures, and LDL- and HDL-subfractions by the Lipoprint system. Basic biochemical parameters including lipid profile were compared with levels in 20 healthy children and were in the physiological range. Improvement of symptoms in the group supplemented with a fish oil emulsion rich in omega-3 FA in contrast to omega-6 FA (emulsion of sunflower oil) has been observed. We are the first to report that omega-3 FAs, but not omega-6 FA, increase large HDL subfractions (anti-atherogenic) after 12 weeks of supplementation and decrease small HDL subfractions (proatherogenic) in depressed children. We found a negative correlation between CDI score and HDL-CH and the large HDL subfraction, but not LDL-CH subfractions. CDI score was not associated with erythrocyte membrane fluidity. Our results suggest that HDL-CH and its subfractions, but not LDL-CH may play a role in the pathophysiology of depressive disorder. The study was registered under ISRCTN81655012.

## 1. Introduction

Depressive disorder (DD) is a prevalent psychiatric disorder that is assumed as the second leading cause of disability. In Europe, the prevalence of depressive disorder in adults is 12% [1] and 5.7% among 13–18 years old with a female to male ratio of 1.3:1 [2].

Symptoms of mood disorders in children include a pervasive and persistent sadness, irritability, decreased school performance, loss of interest and pleasure in social contacts, attention deficit, sleep problems, loss of appetite, and suicidal tendency [3]. Optimal management of child and adolescent depressive disorder includes pharmacotherapy and educational-supportive psychotherapy. The molecular basis of depressive and anxiety disorders in children is not fully understood [4]. It is believed that the establishment and development of depressive disorder involve, among others, nutritional factors contributing through their composition [5].

The increased incidence of DD in people of Western countries has been associated with drastic changes in dietary habits over the century in which the consumption of omega-3 fatty acids (FA) in the form of fish, grain, and vegetables has been replaced by the omega-6 FA from cereal oils. The ratio of omega-3 FA to omega-6 FA in the diet has shifted from 1:1 to 1:15–20, and this switch has coincided with a strong rise in the rates of depression in recent decades [6]. This has led to the hypothesis that omega-3 FA supplementation could represent an approach for treating depression and other mood disorders [7,8,9]. This assumption is supported by the knowledge about the biological effects of omega-3 and omega-6 FA (anti-inflammatory and proinflammatory eicosanoids formation, cell membrane fluidity, cell membrane function, and regulation of proinflammatory cytokine gene expression) [10,11].

The major lipid that plays a role in brain functions is cholesterol. Cholesterol cannot cross the blood–brain barrier and is synthesized locally, namely, by astrocytes and oligodendrocytes. In the circulation, cholesterol binds to apolipoproteins to form various lipoproteins. However, the relationship between circulating cholesterol and its content and function in the central nervous system is not quite known. Cholesterol in the membrane affects its fluidity, which has an impact on the regulation of membrane proteins, transport channels, and synaptic transmission [12].

Several studies have shown a relationship between low levels [13], others with high levels [14] of total cholesterol and depression [15]. Conflicting results are also related to the association of LDL- and HDL-cholesterol with depression. Some studies show a link between depressive disorder and decreased levels of HDL-cholesterol and its apolipoprotein A alongside with increased levels of LDL-cholesterol and its apolipoprotein B [12]. On the other hand, in the meta-analysis of Persons and Fiedorowicz (2016) [16], authors report a negative correlation between depression and LDL-cholesterol. However, these relationships were reported for adults.

The LDL- and HDL-cholesterol lipoprotein fractions are not homogeneous but consist of several subfractions of different size and density. Depending on the method used, LDL can be divided into seven subfractions and HDL into 10 subfractions [17,18]. In general, LDL has been considered to be proatherogenic and HDL lipoproteins to be anti-atherogenic. However, there is no doubt today, that there are both non-atherogenic and atherogenic subfractions in both types of lipoprotein fractions. Large subfractions are currently considered non-atherogenic, even in LDL lipoproteins (IDL-3 and LDL-1) and small subfractions are considered atherogenic, even in HDL lipoproteins (S-HDL8–S-HDL10). A debatable function among LDL subfractions is the LDL2 subfraction, which in adult patients with a non-atherogenic phenotype shows a positive correlation with atherogenic subfractions S-HDL8-10 [19]. On the other hand, hypercholesterolemic adults with a non-atherogenic phenotype had higher levels of LDL2 subfraction compared to controls, and authors concluded that LDL1 and LDL2 do not fulfil the criteria of atherogenicity [18]. The biological function of intermediate subfractions I-HDL4–I-HDL7 is unclear. Oravec et al. (2011) [20] consider intermediate HDL (I-HDL4-7) subfractions as anti-atherogenic part of HDL family. Based on our previous results, we hypothesize that I-HDL4 and I-HDL-5, which are adjacent to large HDL subfractions, are rather non-atherogenic and I-HDL6 a 7, which are adjacent to small HDL subfractions are rather atherogenic [21].

However, the functions of individual subfractions in the pathophysiology of depression are unknown. Cardiovascular diseases (CVDs) have some pathophysiological features similar to those seen in depression [11]. Our previous work shows that administration of omega-3 FA emulsion together with plant sterols and vitamins B6 and B12 to hypercholesterolemic children for 16 weeks significantly reduced serum total cholesterol, LDL- and VLDL-cholesterol and atherogenic subfractions of IDL-1 and IDL-2 [22]. HDL-cholesterol was not changed after supplementation, but in the spectrum of HDL subfractions, there was a shift in the direction of the increase of non-atherogenic L-HDL subfractions and decrease of atherogenic HDL subfractions [23]. Paraoxonase 1 (PON1) is a calcium-dependent glycoprotein esterase, a dimer which is an HDL-associated protein [24]. PON1 has a key role in LDL protection against oxidative damage through breaking down proinflammatory oxidized lipids present in oxidized LDL particles [25]. The appropriate physiological substrate for PON1 activity assessment is unclear. PON1 displays an appreciable arylesterase activity (PON-A) towards phenylacetate and a lactonase activity (PON-L) towards homocysteine-thiolactone [26,27,28]. Relationships between LDL- and HDL-subfractions and PONs activities in hypercholesterolemic children are initially discussed by Muchová et al. (2016) [21]. No significant association between PON1-A and the HDL subfractions in 27 mildly hypercholesterolemic children was found. In contrast, PON1-L activity positively correlated with the antiatherogenic large HDL1 subfraction and negatively with intermediate HDL subfractions 4, 5, and 6 [21].

Whether similar associations can be expected in depressed children is still unanswered. The relationships between lipid markers and depression in children are little known. In addition to cholesterol, polyunsaturated fatty acids (PUFAs) are also associated with depressive disorder. Fifteen to twenty percent of lipids in the brain are omega-3-FA, docosahexaenoic acid (DHA), which is strongly associated with depressive disorder through dysregulation of inflammatory responses, decreased antioxidant capacity, and disordered neurotransmission [29]. Low levels of another omega-3-FA, eicosapentaenoic acid (EPA), which has anti-inflammatory effects and is a competitor to arachidonic acid (omega-6 FA) when incorporated into the membrane, also contribute to depressive pathology. Proinflammatory eicosanoids are formed from arachidonic acid [30]. The incorrect omega-6/omega-3 ratio is thought to contribute to the pathophysiology of depressive disorder rather than the reduced EPA concentration alone [9]. The effect of omega-3 FA on depressive symptoms is thought to exert through membrane fluidity modulation, neuronal development, inflammation, and neurotransmission [31,32]. This study aimed to investigate the relationship between depressive disorder symptoms and lipid profile, LDL-, and HDL-cholesterol subfractions, PON1 activities and erythrocyte membrane fluidity in children and adolescents, as well as the effect of omega-3 FA on the monitored parameters in a double blind, randomized, comparator-controlled study.

## 2. Methods

The presented results are part of the Depoxin project registered under ID number ISRCTN81655012, in which the primary objective was to monitor the effect of omega-3 FA compared to an omega-6 FA on the severity of depression as assessed by the Children’s Depression Inventory (CDI) score [9]. The secondary objective was to monitor the effect of omega-3 FA on selected biochemical parameters, among others, on the lipid profile, lipoprotein subfractions, and interactions between them.

### 2.1. Subjects

A total of 80 children and adolescents participated in this prospective study. From these, sixty out-patients (12 boys and 48 girls) suffering from depressive disorder (DD) (*n* = 31) or mixed anxiety and depressive disorder (MADD) (*n* = 29) registered at the Department of Pediatric Psychiatry of the Faculty of Medicine of Comenius University and The National Institute of Children’s Diseases in Bratislava between June 2013 and December 2018 were enrolled.

Inclusion criteria included the diagnosis of depressive disorder or mixed anxiety and depressive disorder, age 7–18 years, normal eating habits, and no indication of a chronic somatic disease. The diagnoses were determined according to the International Classification of Diseases, 10th edition (ICD 10).

Exclusion criteria were chronic somatic diseases (endocrine, metabolic, and autoimmune), dietary restrictions (vegetarians, lactose intolerance, and celiac disease), psychotic disorders, eating disorders, addiction to psychoactive compounds, personality disorders, organic mental disorders, and pervasive developmental disorders.

All patients, and parents of those who were managed at the National Institute of Children’s Diseases and met diagnostic criteria of depressive disorder, were informed about the possibility to take part in the current trial. Ninety-six patients and their parents were addressed and 60 patients met the inclusion criteria and agreed to take part in the study (Appendix A).

To obtain reference values for the biochemical parameters monitored during the study, twenty healthy children and adolescents (8 boys and 12 girls) registered in the Pediatric Centre Juvenalia, s.r.o Dunajská Streda, Slovakia with an average age of 14.5 ± 2.8 years (11–17 years) were enrolled as the control group of the study.

The study was approved by the Ethics Committee of the National Institute of Children’s Diseases and the Faculty of Medicine, Comenius University Bratislava (20/03/2013). The human study has been performed in accordance with the ethical standards laid down in the 1975 Declaration of Helsinki and its later amendments (2013).

Written informed consent was obtained from parents or legal guardians and children gave verbal assent before participation in the study. Details that might disclose the identity of the subject under study were omitted.

### 2.2. Study Design and Intervention

Patients were randomized to receive either an omega-3 FA rich fish oil emulsion (Om3 group) or a comparator of omega-6 FA rich sunflower oil emulsion (Om6 group) for 12 weeks followed by a wash-out period (4 weeks). Children were included in the study according to ICD 10 with the diagnosis of depressive disorder (DD, *n* = 31; 51.7%) or mixed anxiety and depressive disorder (MADD, *n* = 29; 48.3%).

Simultaneously with their standard antidepressant therapy, children received daily either 20 mL of omega-3 fish oil emulsion (Cultech Ltd., Port Talbot, UK) (providing 2400 mg of total omega-3 FA; 1000 mg EPA and 750 mg DHA, EPA:DHA ratio = 1.33:1) or a comparator of similar appearance—omega-6 sunflower oil emulsion containing 2467 mg of omega-6 linoleic acid (Cultech Ltd., Port Talbot, UK). The dose of omega-3 FA used was determined based on a review of the literature.

Compliance with the product was assessed by monitoring the volume of intervention oil returned and was above 95%.

### 2.3. Randomisation

Trial participants were allocated in a 1:1 ratio to the two arms (Om3 and Om6) according to a computer-generated random sequence using block randomization with a block size of four. The randomization was performed by an independent statistician. Patients were enrolled and assigned sequentially to supplement interventions by the physician. The allocation sequence was not available to any member of the research team until the databases had been completed and locked.

### 2.4. Enrolment and Baseline Characteristics

Patient characteristics such as age (15.7 ± 1.6 years, 11–17 years), gender, weight, height, and body mass index (BMI) (weight (kg)/(height (m)^2^) are shown in the Table 1.

Of the 60 patients included in our study, two dropped out at an early stage (after one to two days after the enrolment); one patient from the Omega-3 group due to the product palatability and one from the Om6 group for the non-compliance (reluctance to miss school every two weeks in order to visit the clinic, difficulties with transportation of outside city patient).

Of the 58 patients who completed the intervention, 29 patients (21 F and 8 M) were included for the data analysis in Om3 group and 29 patients in Om6 group (25 F and 4 M).

Improvement in depressive symptoms was rated as CDI. In parallel, all randomized subjects were analyzed according to randomization.

Patients consumed a standard diet and were advised to consult or report any change in the diet during the intervention.

Both emulsions were well tolerated, and no serious adverse side effects were recorded. Only one patient from the Om3 group stated more frequent defecation (2 to 3 times a day).

The number of patients who left the study, the proportion of diagnoses is shown in the Flow Diagram (Appendix A).

### 2.5. Clinical Investigation

Patient characteristics (age, gender, and menstruation in females) and relevant clinical variables (treatment history—duration of the disease/firstly diagnosed, treatment/no treatment, current medication with antidepressants) were recorded for each patient.

Clinical examinations of all participants were implemented as follows; at the beginning of the trial (week 0) and every 2 weeks for 3 months (weeks 2, 4, 6, 8, 10, 12). The process of data collection is graphically depicted in a Consort flow diagram (Appendix A).

Only the data from patients who completed 12-weeks of intervention were analyzed.

Ratings of depressive severity were made using the self-rated scale of Children’s Depression Inventory (CDI) [33,34] with a higher CDI score representing a more serious depressive state. 

### 2.6. Biochemical Parameters

Venous blood samples were collected from patients and controls after 12-h overnight fast. Within 1 h of collection, blood was centrifuged (1200× *g* 10 min), and serum and plasma (EDTA as an anticoagulant) were obtained and frozen at −80 °C until analysis.

The fasting serum levels of glucose (Glu), creatinine (Crea), uric acid (UA), triacylglycerols (TAG), total cholesterol (TCH), LDL cholesterol (LDL-CH), HDL cholesterol (HDL-CH), urea, creatinine in urine (Crea-U), liver enzymes alanine aminotransferase (ALT), aspartate aminotransferase (AST), gamma-glutamyltransferase (GMT), and high sensitive C reactive protein (hsCRP) were determined at the Department of Clinical Biochemistry of National Institute of Children’s Diseases using a Hitachi 911 Analyzer by a standard procedure using Roche Diagnostics kits to determine the individual parameters (Burgess Hill, RH159RY, UK). 

The analysis of the LDL subfractions in serum was performed by the Lipoprint LDL system (Quantimetrix Corp., Redondo Beach, CA 90278, USA) using polyacrylamide gel electrophoresis [35]. LDL subfractions as well as lipid variables (VLDL, TCH, LDL, and HDL) are expressed as mg/dL.

The analysis of the HDL subfractions in serum was performed by the Lipoprint HDL system (Quantimetrix Corp., Redondo Beach, CA 90278, USA) [17] on polyacrylamide gel electrophoresis. HDL subfractions according to the Lipoprint system are expressed in % of the area under the curve and in mg/dL for the sum of the large, intermediate, and small HDL subfractions.

Arylesterase activity of PON1 (PON-A): For the spectrophotometric determination of PON-A activity in serum, phenyl acetate was used as the substrate according to Gan et al. (1991) [27]. Briefly, the addition of 100 μL of 10 mmol/L substrate solution to the assay mixture containing 10 μL of diluted serum and 890 μL Tris buffer pH 8 (the total volume of the reaction mixture was 1 mL), production of phenol was detected after 2 min at 270 nm. Spontaneous hydrolysis of phenyl acetate was negligible. The molar extinction coefficient 1310 mol^−1^.dm^3^.cm^−1^ was used to express the enzyme activity (U/mL). 1 U is defined as 1 µmol of phenol produced per minute.

Lactonase activity of PON1 (PON-L): For the spectrophotometric measurement of PON1-L activity in serum, dihydrocoumarin was used as the substrate. Lactonase activity was determined according to Aviram and Rosenblat (2008) [26]. After addition of 500 μL of 2 mmol/L substrate solution to the assay mixture containing 15 μL of diluted serum and 485 μL of Tris buffer pH 7.5 (the total volume of the reaction mixture was 1 mL), the absorbance was measured within 4 min at 270 nm.

For calculation of PON1 activity corrections for spontaneous hydrolysis of the substrate were done. The molar extinction coefficient 1295 mol^−1^.dm^3^.cm^−1^ was used to express the enzyme activity (U/mL). 1 U is defined as 1 µmol of dihydrocoumarin hydrolyzed per minute.

### 2.7. Membrane Fluidity

The physical state characteristics of the lipid bilayer of erythrocyte membranes were determined by a fluorescence spectroscopy method and quantified using the anisotropy of the fluorescence probe 1,6-diphenyl-1,3,5-hexatriene (DPH), which is sensitive to the movement of acyl chains of the membrane phospholipids [36]. A high degree of fluorescence anisotropy of DPH (*r_s_*) indicates a high degree of structural order of the phospholipids, which means lower membrane fluidity. Fresh blood samples were centrifuged to remove the plasma and buffy coat. The erythrocytes were washed three times in the cold 0.15 mol/L NaCl, pH 7.4. Prior to fluidity measurements, the erythrocytes were diluted to the hematocrit of 0.04–0.05% to avoid depolarization effects due to light scattering [37].

The suspension of the erythrocytes was labeled with a DPH probe (1,6-diphenyl-1,3,5-hexatriene; Sigma-Aldrich (St. Louis, MO 63178, USA). The samples were excited at 360 nm, the emission was recorded at 443 nm to eliminate any elastically scattered excitation light or energy transfer from DPH to hemoglobin, and the time dependence of *r* was measured for 30 min.

### 2.8. Statistics

Descriptive and univariate analyses were performed on all selected patients’ characteristics. Mean ± standard deviation (SD) is given for the normally distributed variables or a median and interquartile range for data showing departures from normality. Symmetrical data were analyzed with the Student’s *t*-test for independent samples. If the data was skewed but other criteria were met, a nonparametric Mann–Whitney U test was used. In our previous paper [9], we have demonstrated that improvement of clinical symptoms of depression occurred after 6 weeks of supplementation with omega-3 fatty acid. Thus, we have selected the time interval of week 6 to investigate the consequences of omega-3 and omega-6 fatty acid supplementation. To check the effects of omega-3 fatty acid on selected parameters after 6 weeks of supplementation, the repeated measures general linear model (GLM) with the within-subject factor time (week 0 vs. week 6), the between-subject factor group (Om3 vs. Om6) and diagnosis and sex as covariates was used. Tukey pairwise comparisons for the interaction effect were performed. Power of correlations was determined through Spearman rank correlation coefficient.

A value *p* < 0.05 was considered significant in all statistical analyses. For the statistical analysis we employed the statistical programs StatsDirect^®^ 2.8.0 (StatsDirect Sales, Sale, Cheshire, M33 3UY, UK) and Statistica 7 (Statsoft Inc, Tulsa, OK 74104, USA). Graphical representation of data was made using the program GraphPad Prism.

## 3. Results

Our results of the primary outcome of the study—the effect of omega-3 FA on the severity of depressive disorder symptoms—have recently been published by Trebatická et al. (2020) [9]. Significant reductions in CDI scores were observed after 6 and 12 weeks of intervention in the Om3 group compared to the Om6 group. The ratio of omega-6/omega-3 decreased in Om3 but not in Om6 from 24.2/1 to 7.6/1 after 6 weeks; EPA and omega-6/omega-3 ratio, but not DHA, correlated with severity symptoms at the baseline [9].

### 3.1. Baseline Data

The baseline anthropometric data of patients with depression and controls are included in Table 1.

The baseline biochemical parameters (glucose; creatinine; uric acid; lipid profile; liver enzymes such as AST, ALT, GMT, and hsCRP; and creatinine in urine) in the group of patients with depression were initially in the range of physiological values and did not differ from the control group. Omega-3 fatty acid supplementation (Om3 group) was associated with an increase in ALT after 6 weeks and TCH and LDL-CH after 12 weeks. After 6 weeks of supplementation with omega-6 FA (Om6 group), HDL-CH increased. However, the elevated values in both groups were in the range of physiological values (Appendix A) and we do not assume that the increase of parameters would be of fundamental pathophysiological significance.

As the effect of total cholesterol on mental status is discussed, we divided patients into two groups depending on TCH levels and monitored the CDI score. No difference in CDI score dependent on TCH level was found in our children and adolescents (25.7 CDI score in the group with TCH < 4.0, *n* = 36 and 25.5 score in the group with TCH > 4.0, *n* = 22). The correlation between CDI score and TCH in the stratified groups showed a trend towards a negative correlation in the group with TCH < 4.0 (*r* = −0.216, *p* = 0.09) and a trend towards a positive correlation in the group with TCH > 4.0 (*r* = 0.315, *p* = 0.076), whereas 44% of patients with MADD were in the group with TCH < 4.0 and 56% of patients with MADD in the group with TCH > 4.0.

### 3.2. Lipoprotein Subfractions

LDL-lipoprotein subfractions and VLDL were analyzed in the serum of patients and controls. Patients had increased atherogenic VLDL by 35% and decreased atherogenic IDL-2 (by 23%), non-atherogenic LDL-1 (by 16%), and supposedly non-atherogenic LDL-2 (by 18%) subfractions compared to controls (Figure 1, Appendix A).

Supplementation with omega-3 FAs increased non-atherogenic LDL-1 (by 13%) and LDL-2 (by 39%) subfractions after 6 weeks and LDL-1 (by 12%) after 12 weeks (Appendix A; Figure 2). Omega-6 FAs similarly increased the non-atherogenic subfractions of LDL-1 (by 10%) and LDL-2 (by 47%). However, after 12 weeks of omega-6 FA supplementation, an increase in the atherogenic IDL-2 subfraction (by 10%) was also observed (Appendix A; Figure 3). However, significant differences in LDL subfractions between Om3 and Om6 groups were not confirmed (Appendix A).

From HDL-lipoprotein subfractions we found higher levels of anti-atherogenic subfraction L-HDL1 (by 49%) and lower supposedly anti-atherogenic subfractions I-HDL4 and I-HDL-5 (by 8% and 10%, respectively) compared to controls (Appendix A; Figure 4).

Supplementation with omega-3 FAs increased non-atherogenic large HDL-subfractions (L-HDL1, L-HDL2, L-HDL3, and I-HDL4) by 22%, 16%, 19%, and 7%, respectively, and after 12 weeks increased large HDL subfractions by 19%, 11%, and 13%. Atherogenic HDL subfractions (I-HDL6, I-HDL7 decreased by 12% and 15%, after 12 weeks by 8%, 10%, and S-HDL8 by 11% (Appendix A; Figure 5). Omega-6 FA supplementation did not affect HDL subfraction levels (Appendix A; Figure 6).

Basal values of all HDL subfractions did not differ significantly in the Om3 and Om6 groups (Appendix A). Statistical analysis of HDL subfractions after supplementation with omega-3 and omega-6 FAs showed a significant difference in the level of L-HDL2 (*p* = 0.011) subfraction and a marginally significant difference in L-HDL1 (*p* = 0.084) and L-HDL3 (*p* = 0.070) subfractions after 6 weeks of supplementation as well as S-HDL8 (*p* =0.078) subfraction after 12 weeks of supplementation between the Om3 and Om6 groups (Appendix A).

The repeated measures GLM revealed a significant time*group (time*treatment) interaction on non-atherogenic large HDL subfractions L-HDL1 (F_(1, 54)_ = 4.89, *p* = 0.031), L-HDL2 (F_(1, 54)_ = 4.88, *p* = 0.032), and L-HDL3 (F_(1, 54)_ = 9.03, *p* = 0.004) levels. Tukey post hoc comparisons showed that levels of these large HDL subfractions were significantly increased in the group of children after 6 weeks of supplementation with omega-3 FA (for L-HDL1: *p* = 0.021, L-HDL2: *p* = 0.007, and L-HDL3: *p* = 0.001), but not in the group of children with omega-6 FA supplementation. The six-week supplementation with omega-3 FAs increased levels of L-HDL1, L-HDL2, and L-HDL3 by 22%, 16%, and 19%, respectively (Appendix A, Figure 5). GLM showed that sex and diagnosis were not significant covariates.

There was also a significant time*group (time*treatment) interaction on atherogenic L-HDL6 (F_(1, 54)_ = 6.72, *p* = 0.012) and L-HDL7 (F_(1, 54)_ = 4.51, *p* = 0.038) levels. Tukey pairwise comparisons revealed that L-HDL6 and L-HDL7 levels were significantly decreased in the group of children after 6 weeks of supplementation with omega-3 FA (*p* = 0.001 and *p* = 0.003, respectively), but not in the group of children with omega-6 FA supplementation. The six-week supplementation with omega-3 FAs lowered levels of L-HDL6 and L-HDL7 by 13% and 15%, respectively (Appendix A, Figure 5). Supplementation with omega-6 FA had no significant effect on the level of L-HDL6 and L-HDL7 subfractions (Appendix A, Figure 6). GLM showed that sex and diagnosis were not significant covariates.

However, it should be noted that the presented results of the effect of omega-3 FA supplementation on HDL subfractions in depressed children are only pioneering results and were obtained in a small number of patients and therefore cannot be generalized.

Other parameters, such as TCH, HDL, TAG, LDL, LDL1, LDL2, and I-HDL4 did not significantly differ between the Om3 and Om6 groups after 6 and 12 weeks of supplementation with omega-3 or omega 6 FA as shown by repeated measures GLM.

### 3.3. Paraoxonase Activity

We investigated the activity of the enzyme paraoxonase 1 (PON) with lactonase (PON-L) and arylesterase (PON-A) activities. We did not find a difference in both PONs activities between patients and healthy controls, as well as in the activities after 6 and 12 weeks of omega-3 and omega-6 FA supplementation (Appendix A).

Further, we were interested in correlations between the severity of depression expressed as a CDI score and selected parameters of the lipid profile and individual LDL- and HDL-subfractions at the week 0 and week 6 of investigation (Table 2). Similarly, the correlations between selected HDL subfractions and activities of PON-L and PON-A are listed in Table 2. Correlations between the activity of PON-L or PON-A and CDI or HDL at the baseline and after supplementation with omega-3 FA (group Om3) and omega-6 FA (group Om6) are listed in Table 3.

### 3.4. Fluidity of Erythrocyte Membranes

We have hypothesized that the supplementation with omega-3/omega-6 FAs would influence the fatty acid composition of membrane lipids, which would lead to increased fluidity and reduced surface packing density of the membrane component. To assess the presumable effects, we determined membrane fluidity in the isolated erythrocytes—a simple, easily obtainable model of membrane phospholipid dynamics. The extent of the viscosity (or its reciprocal, fluidity) was quantified in terms of the steady-state anisotropy of a fluorescent dye DPH (*r_s_*). We did not observe any difference between controls and depressed children (*r_s_* = 0.241 vs. 0.244, *p* = 0.561). Supplementation with FA decreased anisotropy (increased fluidity) compared to the baseline after 6 and 12 weeks of supplementation with both, omega-3 (from *r_s_* = 0.246 to 0.228 and 0.22, *p* < 0.0001) and omega-6 FA (from *r_s_* = 0.241 to 0.23 and 0.22, *p* < 0.0001).

Correlation analysis between anisotropy and lipid parameters showed a significant positive relationship between anisotropy and TCH in the patients (*r_s_* = −0.317, *p* = 0.038), but not in the controls. We have found a weak positive correlation between anisotropy and LDL-CH at the baseline (*r_s_* = −0.277, *p* = 0.062). The *p*-value was indicative of a borderline significance.

Correlation between anisotropy and CDI score was not observed at any time of the investigation.

## 4. Discussion

In the present study, 58 patients with depressive disorder and 20 healthy controls were assessed and compared for basic biochemical parameters including TCH, LDL-, and HDL-cholesterol, and TAG. All evaluated parameters were in the physiological range without a difference between patients and controls. Omega-3-FA supplementation caused an increase in TCH and LD1-CH after 12 weeks, and omega-6-FA caused an increase in HDL-CH after 6 weeks, but this increase was within physiological values for all altered parameters. Unlike our depressed children, Enko et al. (2018) [15] found that depressed adults had significantly increased plasma TAGs and decreased HDL-CH compared to controls. However, the results of the study of the relationships between depression and lipid parameters are controversial. In one study, depression in adults was associated with higher TAG, TCH, and LDL-CH and in another study lower HDL-CH [38,39] compared to the controls. In other studies TAG was unchanged and lower TCH and LDL-CH ([40], or higher TAG and HDL-CH [41] were found in patients compared to healthy individuals. Differences in studies are attributed to considerable heterogeneity of cohorts. However, these results have been found in adults. The association of individual lipid parameters with the pathology of depression in children and adolescents is less known. Finnish longitudinal study investigated the association between serum lipids (TAG, TCH, LDL-, and HDL-cholesterol) and depressive symptoms during one year follow-up (from 3 to 9 years at the baseline). Results indicated that a rapid increase of TAG at an early age, but not other lipids, were predictive of depression onset [42].

However, lipid parameters (TCH, LDL-, HDL-cholesterol, and TAG) and BMI in our cohort of depressed children and adolescents were not significantly different at the baseline from healthy controls.

The importance of cholesterol in the function of the neural system has been underlined [42]. Low levels of cholesterol in the membrane are thought to reduce the number of serotonin receptors and thus reduce the level of serotonin in the brain [43]. On the other hand, elevated cholesterol levels may contribute to the altered fluidity of the neuronal membrane. It seems that patients with a low cholesterol level (<4.0 mmol/L) are at an increased risk of suicide, while patients with an elevated cholesterol level (>5.6 mmol/L) more often exhibit resistance to treatment and more frequently have a comorbid anxiety disorder [44]. The results so far show that both elevated and decreased cholesterol levels may be related to depression. Whether the opposite trend of correlation between CDI score and TCH level could be influenced by the different proportion of patients with MADD (additional comorbidity) cannot be generalized to a small number of patients in both groups.

However, our results are consistent with the findings of Martinac et al. (2007) [44] and support the opinion that the relationship between lipid parameters and depressive disorder may also be dependent on other psychiatric comorbidities.

Unlike adults, our depressed children and adolescents did not have significantly different TCH, LDL-, and HDL-cholesterol levels from the control group. However, HDL-CH, but not LDL-CH, inversely correlated with CDI score in contrast to results followed from meta-analyses in adults, where increased HDL-CH positively correlated with depression, especially in women [45]. However, we did not evaluate gender correlations for the small number of boys enrolled in the study.

The atherogenic lipoprotein profile is characterized by the presence of VLDL, IDL-1 (intermediate density lipoproteins), IDL-2, and small dense LDL-3 to LDL-7 subfractions in LDL subfractions [23]. IDL-3, LDL-1, and suspiciously LDL-2 do not meet the criteria for atherogenesis [18]. Conversely, in the spectrum of HDL subfractions, small HDL subfractions (S-HDL8-10) and intermediate HDL subfractions (I-HDL6-7) are considered atherogenic and large HDL subfractions (L-HDL1-3) and suspiciously I-HDL4-5 are considered non-atherogenic fractions [18,20,21].

Our results of lipid parameters and lipoprotein subfractions show that children with depressive disorder do not show a potential risk for the development of later CVDs.

At the baseline LDL-CH did not differ between the patients and controls. Even though non-atherogenic LDL1 subfraction increased after both omega-3 and omega-6 FA supplementations, none of the LDL subfractions showed any correlation with CDI scores at baseline or after supplementation with omega-3 and omega-6 FAs. As neither LDL-CH nor LDL subfractions correlated with CDI scores, the predicted association with depressive disorder in children and adolescents seems debatable.

In our study, omega-3 FA supplementation significantly affected the levels of more HDL subfractions, in contrast to LDL subfractions, after 6 and 12 weeks of supplementation.

As the basal values of individual subfractions in the Om3 and Om6 groups were similar, the effects of omega-3 and omega-6 FA supplementation on HDL subfractions was evaluated using pre–post comparison (week 0 vs. weeks 6 and 12) in both groups as well by statistical analysis of a significant difference between the Om3 and Om6 groups in individual HDL subfractions. Significant differences were found for large HDL subfractions, L-HDL2 and the sum of large HDLs, L-HDL1+2+3. The marginally significant differences between Om3 and Om6 in L-HDL1 and L-HDL3 should be explained by the inter-variability and the small number of included individuals in the given groups.

To our knowledge, the relationships between depression severity and lipoprotein subfractions have not been studied to date. Therefore, it is not possible to confront our results with the results of other authors. In our previous work, in which hypercholesterolemic children were supplemented with an emulsion of fish oil enriched with omega-3 FA together with plant sterols and vitamins B6 and B12 for 16 weeks we found a shift in the direction of the increase of non-atherogenic L-HDL subfractions and decrease of atherogenic HDL subfractions [23]. Similarly, in our current work, supplementation with omega-3 FAs, but not omega-6 FAs, shifted the spectrum of HDL subfractions in depressed children in favor of large HDL subfractions (L-HDL1 to 3) after 6 and 12 weeks of supplementation and borderline significant reduced level of small HDL subfraction (S-HDL8) after 12 weeks of supplementation.

Although the serum HDL-CH value in patients was not significantly different from the serum value of HDL-CH in healthy controls, a negative correlation of HDL-CH with CDI score and of non-atherogenic L-HDL-3 subfraction with a CDI score as well as a positive correlation of supposedly atherogenic I-HDL6 subfraction with CDI before omega-3 FA supplementation points to a hypothesis of a potential role of HDL subfractions in the pathophysiology of depressive disorder.

Paraoxonase 1 is an enzyme associated with HDL lipoproteins. We also determined activities of paraoxonase1 against both substrates. We did not observe any differences in PON-A and PON-L activities between patients and controls. Similarly, both supplements (omega-3 FA and omega-6 FA) did not affect baseline enzyme activities. Kodydkova et al. (2009) [46] found that the activity of PON1 in adult women with depressive disorder was not significantly different from healthy controls. However, in another work [47] authors found, that major depression, but not bipolar disorder in adults, was accompanied by lower PON1 activity. PON1 activity was decreased by smoking and was diagnosed by genotypic interaction (i.e., lower PON1 in major depression with the QQ genotype). Consistent with the assumption as well as with the results in hypercholesterolemic children [21], in our depressive cohort PON-L, but not PON-A, activity correlated with HDL-CH as well as with the CDI score at the baseline. Neither omega-3 FA nor omega-6 FA supplementations affected CDI correlations with PONs.

Assuming that the preferred distribution of fluorescent DPH molecules is normal to the membrane surface, the dye molecules can detect the movement of the ends of the phospholipid chain in the region of the membrane core [48,49]. Thus, the enhanced disorder associated with the multiple double bonds in DHA is the likely molecular origin for the observed lower anisotropy of the fluorescent probe DPH at the 6 and 12-week supplementation in both groups. It indicates that the membranes had “looser” membrane structure compared to their respective baseline values.

Such a modification in the membrane structure and fluidity, and consequently in the membrane thickness directly affects the degree of exposure of membrane proteins on the cell surface, influences accessibility of reactive chemical groups and induces conformational changes of the proteins. Thus, the interrelated physical and biochemical events in plasma membranes may have an impact on membrane functions, signaling, neurotransmission, permeability, and other cellular events [50,51].

To the best of our knowledge, there is no published work investigating the fluidity of erythrocyte membranes in depressed children. Kališová-Stárková et al. (2006) [52] examined the anisotropy of erythrocyte membranes (fluidity) in 24 adult patients with depressive disorder and found a higher value of anisotropy (*r_DPH_*) (it means lower fluidity) in patients compared to the control group. Patients were taking antidepressant citalopram for one month. However, the treatment did not affect the value of anisotropy and anisotropy did not correlate with depressive symptoms similarly to our results with omega-3 FA supplementation in children and adolescents. In accordance with Kališová-Stárková et al. (2006) [52], we also found a positive correlation of anisotropy with serum TCH. Our results confirmed that the composition of membranes in depressed patients becomes altered but does not correlate directly with depressive symptoms.

Our study has some limitations:

We did not collect the food diaries, however the patients were advised to follow their normal eating habits and not consume any fatty acid supplements or antioxidants during the study period. We included only a small number of boys, therefore the statistical gender evaluation should be interpreted with caution.

## 5. Conclusions

Our results suggest that omega-3 FA, in contrast to omega-6 FA, increases large HDL subfractions and decreases small HDL subfractions, while large HDL subfractions are associated with reduced and small HDL-subfractions with worsening of the severity of depressive symptoms. LDL subfractions are not associated with depressive symptoms. HDL-CH and its subfractions, but not LDL-CH or its subfractions, may play a role in the pathophysiology of depressive disorder in children and adolescents. However, the results of our study are obtained in a relatively small number of depressed children and adolescents and therefore require verification of the findings in a larger number of patients.

## Figures and Tables

**Figure 1 biomolecules-10-01427-f001:**
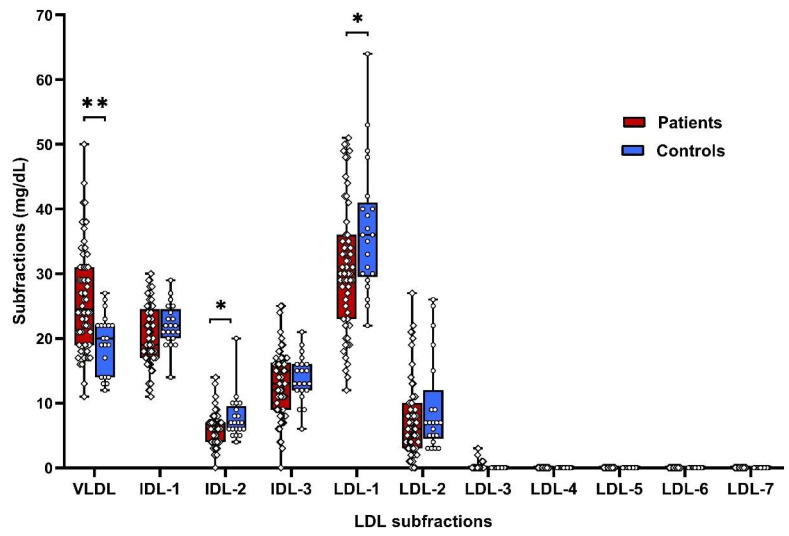
LDL subfractions in controls and patients at the baseline. VLDL—very low-density lipoproteins, IDL—intermediate density lipoproteins, LDL—low density lipoproteins, * *p* < 0.05, ** *p* < 0.01.

**Figure 2 biomolecules-10-01427-f002:**
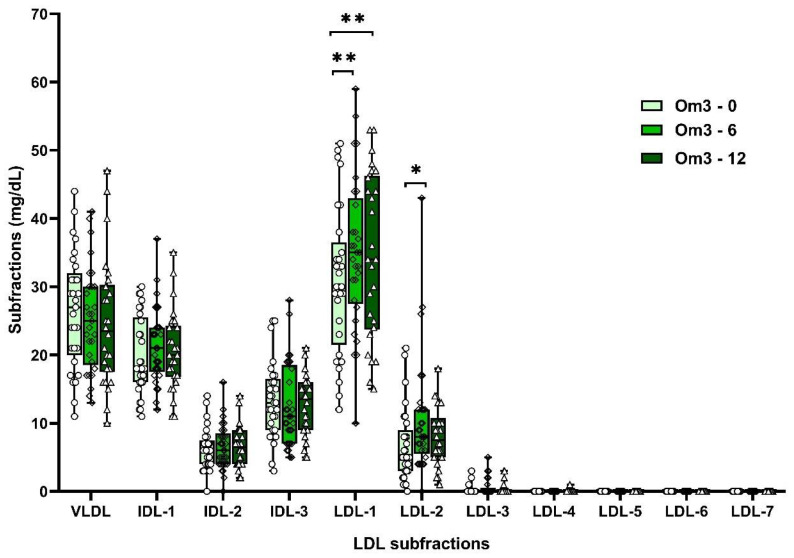
LDL subfractions in Om3 group at the baseline and after 6 and 12 weeks of supplementation. VLDL—very low-density lipoproteins, IDL—intermediate density lipoproteins, LDL—low-density lipoproteins, Om3—omega-3 group, * *p* < 0.05, ** *p* < 0.01.

**Figure 3 biomolecules-10-01427-f003:**
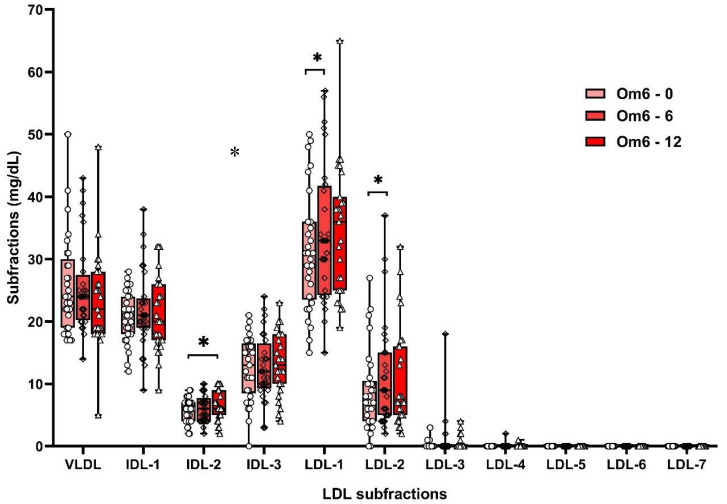
LDL subfractions in Om6 group at the baseline and after 6 and 12 weeks of supplementation. VLDL—very low-density lipoproteins, IDL—intermediate density lipoproteins, LDL—low-density lipoproteins, Om6—omega-6 group, * *p* < 0.05.

**Figure 4 biomolecules-10-01427-f004:**
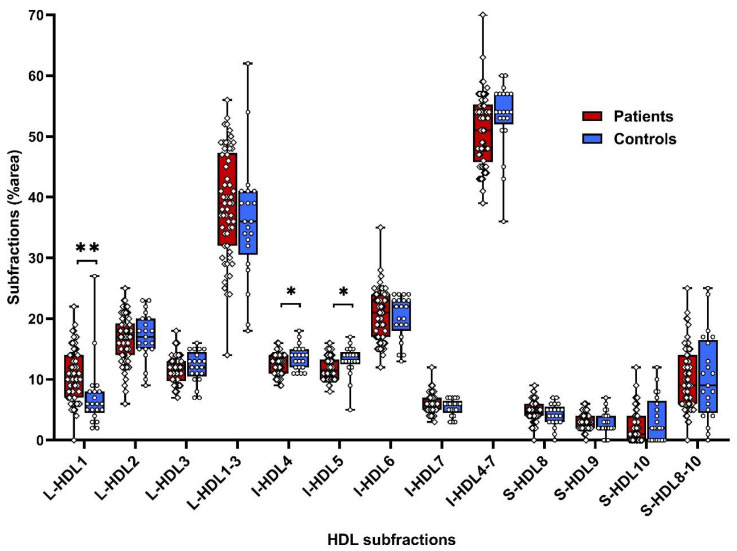
HDL subfractions in controls and patients at the baseline. HDL—high density lipoprotein, L-HDL—large HDL subfraction, I-HDL—intermediate HDL subfraction, S-HDL—small HDL subfraction, * *p* < 0.05, ** *p* < 0.01.

**Figure 5 biomolecules-10-01427-f005:**
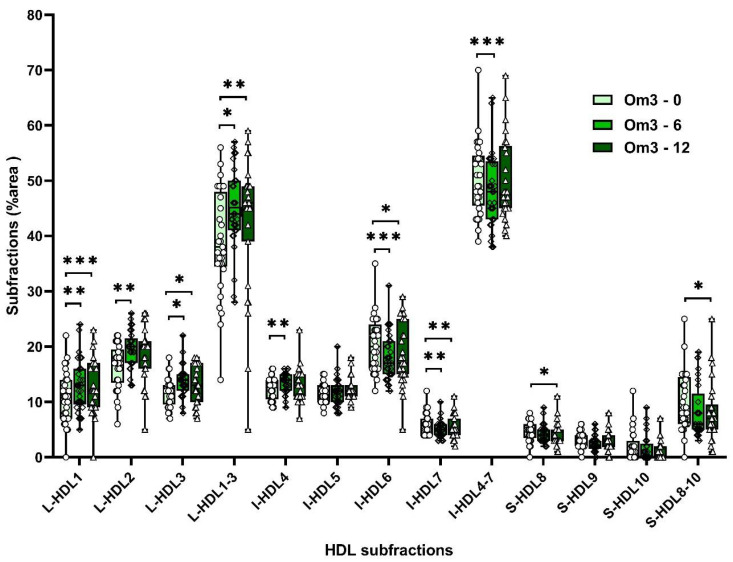
HDL subfractions in Om3 group at the baseline and after 6 and 12 weeks of supplementation. HDL—high-density lipoprotein, L-HDL—large HDL subfraction, I-HDL—intermediate HDL subfraction, S-HDL—small HDL subfraction, Om3—omega 3 group, * *p* < 0.5, ** *p* < 0.01, *** *p* < 0.001.

**Figure 6 biomolecules-10-01427-f006:**
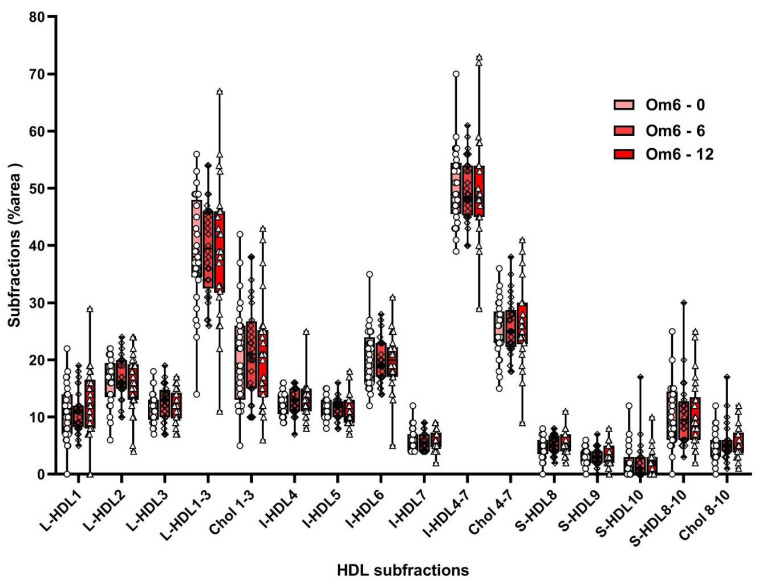
HDL subfractions in Om6 group at the baseline and after 6 and 12 weeks of supplementation. HDL—high-density lipoprotein, L-HDL—large HDL subfraction, I-HDL—intermediate HDL subfraction, S-HDL—small HDL subfraction, Chol—cholesterol, Om3—omega 3 group.

**Table 1 biomolecules-10-01427-t001:** Anthropometric parameters of evaluated depressive patients and controls.

Parameter	Patients			*p*	Controls			*p*	*p*
	All	Male	Female	M vs. F	All	Male	Female	M vs. F	P vs. C
*n*	*n* = 58	*n* = 12	*n* = 46		*n* = 20	*n* = 8	*n* = 12		
Age (years)	15.6 ± 1.6	16.4 ± 2.2	15.3 ± 1.3	0.235	14.8 ± 2.4	14.0 ± 2.5	14.4 ± 2.5	0.621	0.059
Weight (kg)	60.3 ± 11.7	68.2 ± 15.5	56.8 ± 9.3	0.015	54.9 ± 18.8	56.1 ± 21.2	54.1 ± 17.8	0.435	0.147
Height (m)	1.68 ± 0.1	1.74 ± 01	1.66 ± 0.1	0.016	1.6 ± 0.2	1.6 ± 0.2	1.6 ± 0.1	0.543	0.471
BMI (kg/m2)	21.14 ± 2.7	22.4 ± 3.6	20.5 ± 2.8	0.532	20.6 ± 4.2	20.1 ± 3.0	20.9 ± 5.0	0.498	0.537

M—male, F—female, *p*—significance, Om3—omega-3 group, Om6—omega-6 group, vs—versus.

**Table 2 biomolecules-10-01427-t002:** Correlations between CDI score and lipid profile parameters, and LDL- and HDL-subfractions.

**All patients at the week 0**	***n***	***r***	***p***
CDI vs. TCH	58	−0.043	0.374
CDI vs. LDL-CH	58	0.024	0.427
CDI vs. HDL-CH	58	−0.226	**0.043**
CDI vs. TAG	58	0.062	0.321
**All patients at the week 0**	***n***	***r***	***p***
TCH vs. LDL-CH	58	0.882	**0.0001**
TCH vs. HDL-CH	58	0.011	0.467
TCH vs. TAG	58	0.508	**0.0001**
**All patients at the week 0** **LDL-subfractions**	***n***	***r***	***p***
CDI vs. VLDL	58	−0.126	0173
CDI vs. IDL1	58	0.024	0.427
CDI vs. IDL2	58	0.151	0.129
CDI vs. IDL3	58	0.082	0.27
CDI vs. LDL1	58	0.046	0.365
CDI vs. LDL2	58	−0.044	0.372
**All patients at the week 0** **HDL-subfractions**	***n***	***r***	***p***
CDI vs. L-HDL1	58	−0.032	0.405
CDI vs. L-HDL2	58	−0.099	0.23
CDI vs. L-HDL3	58	−0.446	**0.0002**
CDI vs. I-HDL4	58	−0.169	0.102
CDI vs. I-HDL5	58	0.087	0.257
CDI vs. I-HDL6	58	0.363	**0.003**
CDI vs. I-HDL7	58	0.19	0.076
**Om3 group at the week 6** **HDL subfractions**	***n***	***r***	***p***
CDI vs. L-HDL3	28	0.092	0.325
CDI vs. I-HDL6	28	−0.108	0.296
CDI vs. I-HDL7	28	0.073	0.358
PON-L vs. L-HDL3	28	−0.159	0.209
PON-L vs. I-HDL6	28	−0.092	0.321
PON-L vs. I-HDL7	28	0.190	0.167
PON-A vs. L-HDL3	29	−0.082	0.337
PON-A vs. I-HDL6	29	0.029	0.440
PON-A vs. I-HDL7	29	0.133	0.245

CDI—Children’s Depression Inventory, LDL—low density lipoprotein, HDL—high density lipoprotein, LDL-CH—low density lipoprotein cholesterol, HDL-CH—high density lipoprotein cholesterol, TCH—total cholesterol, TAG—triacylglycerol, VLDL—very low density lipoprotein, IDL—intermediate density lipoprotein, L-HDL—large HDL, I-HDL—intermediate HDL, PON-L—paraoxonase 1 with lactonase activity, PON-A—paraoxonase activity with arylesterase activity, Om3—omega-3 group, *n*—number of subjects, *r*—correlation coefficient, *p*—significance

**Table 3 biomolecules-10-01427-t003:** Correlations between activity of PON-L and PON-A vs. CDI and HDL.

**All Patients at the Week 0**	***n***	***r***	***p***
CDI vs. PON-L	58	−0.221	**0.048**
CDI vs. PON-A	58	−0.087	0.257
PON-L vs. HDL	58	0.295	**0.018**
PON-A vs. HDL	58	0.010	0.472
**Om3 Group at the Week 6**	***n***	***r***	***p***
CDI vs. PON-L	28	−0.075	0.353
CDI vs. PON-A	28	−0.304	0.058
CDI vs. HDL	28	0.028	0.446
HDL vs. PON-L	28	0.057	0.388
HDL vs. PON-A	28	−0.013	0.474
**Om6 Group at the Week 6**	***n***	***r***	***p***
CDI vs. PON-L	29	−0.136	0.246
CDI vs. PON-A	28	−0.184	0.175
CDI vs. HDL	28	−0.068	0.369
HDL vs. PON-L	29	0.453	**0.009**
HDL vs. PON-A	27	0.067	0.369

CDI—Children’s Depression Inventory, HDL—high density lipoprotein, L-HDL—large HDL, I-HDL—intermediate HDL, PON-L—paraoxonase 1 with lactonase activity, PON-A—paraoxonase activity with arylesterase activity, Om3—omega-3 group, Om6—omega-6 group, *n*—number of subjects, *r*—correlation coefficient, *p*—significance.

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
