# Peer review of "Lipid Profile, Lipoprotein Subfractions, and Fluidity of Membranes in Children and Adolescents with Depressive Disorder: Effect of Omega-3 Fatty Acids in a Double-Blind Randomized Controlled Study"

_biomolecules, 2020, doi:10.3390/biom10101427_

Round 1

Reviewer 1 Report

No further comments

Author Response

Reviewer No 1:

Comments and Suggestions for Authors :

No further comments

Thanks to Reviewer 1 for his review of our revised manuscript.

Thank you for his contribution to improving our manuscript.

Reviewer 2 Report

It is a well-done duble-blind randomized controlled study on the relationship between depressive disorder symptoms and lipid profile. The paper seems to be a interesting contribution to our knowledge of the pathophysiology of depressive disorder. The methodology and presentation of results is on a high scientific level. The overview of the literature as well as the discussion is modern and comprehensive. The reviewer does not suggest any changes.

Author Response

Answer to Reviewer No 2:

Comments and Suggestions for Authors

It is a well-done double-blind randomized controlled study on the relationship between depressive disorder symptoms and lipid profile. The paper seems to be an interesting contribution to our knowledge of the pathophysiology of depressive disorder. The methodology and presentation of results are on a high scientific level. The overview of the literature as well as the discussion is modern and comprehensive. The reviewer does not suggest any changes.

Thanks to Reviewer 2 for his review of our revised manuscript.

Thank you for his contribution to improving our manuscript.

Reviewer 3 Report

The authors report findings on secondary and exploratory endpoints (related to lipids) from a randomised controlled trial on omega-3 FA vs omega-6 FA supplementation in adolescents with depression.  The paper is well written and structured and covers a timely topic as the existing literature on the relationship between blood lipid levels and psychiatric disorders is still inconsistent. However, I have some comments particularly on the statistical analysis and presentation.

Abstract: should be more structured, please refer to CONSORT guidelines on reporting of randomised trials. Slightly more detail on methods (sample, intervention etc.) and results (p-values, effect sizes) is needed. Clearly state that this paper reports on secondary/exploratory endpoints, not the primary endpoint (depression scores). The main results reported here should focus on the om3 vs om6 comparison, not on pre-post comparisons.

Statistical methods: in such a randomised trial, the focus of analysis should always be the comparison between randomised groups at post-treatment time-points (here week 6 and 12), but not on pre-post comparisons of the groups separately. Additionally, at baseline (week 0 here) both groups come from the same population and any small differences are due to chance (randomisation), so conceptually it does not make sense to test for differences between randomised groups at baseline (e.g. in tables S3, S4). As a consequence, using a repeated measures GLM with week 0 data as part of the dependent variables is not sensible for randomised trials, while it would be OK for just week 6 and week 12 outcome data (and possibly using week 0 as a covariate).

Results: please again focus on the group comparisons (om3 vs. om6), not the pre-post (week 0 vs week 6 or 12) in the separate groups (as e.g. in tables S3, S4 and extensively in the text).

Figures 1 to 6 only show means (as bar plots), but no measure of the variability or whole distribution. I would very much recommend to use box plots (which show much more of the whole distribution) or even box plots overlaid with dot plot (as the sample size is rather small, dots would be nice).

Multiple testing should be critically discussed, as the number of tests performed here is quite large, considering the small sample size. I would not necessarily advise to use e.g. Bonferroni correction as this is quite conservative. But the exploratory character of these analyses needs to be stressed.

Minor points:

Supplementary table S3: p-values for LDL-1 in om6 group seem slightly implausible as the difference between week 0 and week 6 is significant while week 0 to week 12 is not, even though week 12 mean is higher and SD smaller than at week 6 (however, without full data, this is just speculation on my part). Please carefully check this (if these pre-post comparisons are left in).

Line 363-364 should probably refer to table S4, not S3.

Why does table 3 only contain separate results of om3 group, not om6 group?

Lines 451 to 457 in the discussion contain results which should be moved to the result section and only be discussed here.

Line 475 “not different” should be “not significantly different” as of course individual values as well as the point estimates for population means are different, but this absence of a significant difference should not be interpreted as evidence of “no effect” in such a small sample.

Do not report p=0.000 in supplementary table S4 (twice), but rather p<0.001 (as the p-value can never be exactly 0).

Author Response

Comments and Suggestions for Authors – reviewer 3

The authors report findings on secondary and exploratory endpoints (related to lipids) from a randomised controlled trial on omega-3 FA vs omega-6 FA supplementation in adolescents with depression.  The paper is well written and structured and covers a timely topic as the existing literature on the relationship between blood lipid levels and psychiatric disorders is still inconsistent. However, I have some comments particularly on the statistical analysis and presentation.

Abstract: should be more structured, please refer to CONSORT guidelines on reporting of randomised trials. Slightly more detail on methods (sample, intervention etc.) and results (p-values, effect sizes) is needed. Clearly state that this paper reports on secondary/exploratory endpoints, not the primary endpoint (depression scores). The main results reported here should focus on the om3 vs om6 comparison, not on pre-post comparisons.

We would like to satisfy the reviewer regarding the comments on the abstract. However, we were limited by the editorial conditions: "The abstract should be a total of about words maximum:  Our number of words in this form is 211. We have modified the abstract within the framework of the given conditions.to 278 words.

Statistical methods: in such a randomised trial, the focus of analysis should always be the comparison between randomised groups at post-treatment time-points (here week 6 and 12), but not on pre-post comparisons of the groups separately. Additionally, at baseline (week 0 here) both groups come from the same population and any small differences are due to chance (randomisation), so conceptually it does not make sense to test for differences between randomised groups at baseline (e.g. in tables S3, S4). As a consequence, using a repeated measures GLM with week 0 data as part of the dependent variables is not sensible for randomised trials, while it would be OK for just week 6 and week 12 outcome data (and possibly using week 0 as a covariate).

I understand the reviewer's comment. Thanks for his opinion. However, the effect of any substance on a particular symptom is evaluated by comparing the state before and after administration of the substance. Therefore, I don't think it would be right to omit data at time 0.

 In our study, we report the basal values ​​of lipid markers because there are works on changes in basic lipid parameters and the severity of depression. We also compared them in the Om 3 and Om6 groups for another reason (we found their difference to be insignificant in accordance with the assumption of correct randomization). When evaluating the primary outcome (severity of depressive symptoms), we found a difference in the basal values ​​of the CDI score, which evaluates the severity of symptoms, and therefore we used the basal value as a co-variant when evaluating the effect of omega-3. And even in this case, the effect of omega-3 proved to be significant on the symptoms of depression.

In the case of the lipoprotein subfractions, we did not find any differences in basal values ​​between the Om3 and Om6 groups (Table S4).

Of course, the evaluation of the effect is considered to be credible in placebo-controlled studies, in which the effect over time between the active substance and the placebo should be compared. We performed a statistical analysis of the differences in individual parameters between the Om3 and Om6 groups. We found significant differences between Om3 and Om6 groups in L-HDL2 subfractions and borderline significant in subfractions L-HDL1 and L-HDL3 at weeks 6 and 12 and for S-HDL8 after 12 weeks of supplementation (Table S4). This information is given in the text of the manuscript – 22-09-2020, lines 344-363.

Results: please again focus on the group comparisons (om3 vs. om6), not the pre-post (week 0 vs week 6 or 12) in the separate groups (as e.g. in tables S3, S4 and extensively in the text).

The text regarding the effect of supplementation on LDL subfractions states that the difference in the effect of omega-3 and omega-6 FA on individual subfractions was not confirmed (lines 346-351).

The effect of supplementation on HDL subfractions is shown in lines 326-330 and states that a significant difference (increase) was found between the Om3 and Om6 groups in antiatherogenic large HDL subfractions after 6 and 12 weeks of supplementation and in atherogenic small HDL subfractions S-HDL8 and S-HDL10 (decrease) between Om3 and Om6 groups (lines 344-363).

Figures 1 to 6 only show means (as bar plots), but no measure of the variability or whole distribution. I would very much recommend to use box plots (which show much more of the whole distribution) or even box plots overlaid with dot plot (as the sample size is rather small, dots would be nice).

The figures were redesigned according to the reviewer's recommendation.

Multiple testing should be critically discussed, as the number of tests performed here is quite large, considering the small sample size. I would not necessarily advise to use e.g. Bonferroni correction as this is quite conservative. But the exploratory character of these analyses needs to be stressed.

The comment for this is inserted on lines 367-369.

Minor points:

Supplementary table S3: p-values for LDL-1 in om6 group seem slightly implausible as the difference between week 0 and week 6 is significant while week 0 to week 12 is not, even though week 12 mean is higher and SD smaller than at week 6 (however, without full data, this is just speculation on my part). Please carefully check this (if these pre-post comparisons are left in).

We recalculated the averages and statistical data very carefully and did not find any error in the calculations.  A significant difference between weeks 0 and 12 for LDL-1 subfraction in the Om6 group is unlikely due to large scatter (see Fig. 3).

 Line 363-364 should probably refer to table S4, not S3.

Thank you for reporting an error / typo. The text is corrected (line 361-363 an also 355).

Why does table 3 only contain separate results of om3 group, not om6 group?

In the Om6 group, no change was found after supplementation at weeks 6 and 12 (Table S4) in contrast to the Om3 group, while a significant effect of omega-3 FA on subfraction levels was found.  Therefore, in the table of correlations with the CDI score, we show only the Om3 group.

Lines 451 to 457 in the discussion contain results which should be moved to the result section and only be discussed here.

The recommended move of the text to "Results" was done (lines 284-290) and the discussion of these results was added (lines 432-442).

Line 475 “not different” should be “not significantly different” as of course individual values as well as the point estimates for population means are different, but this absence of a significant difference should not be interpreted as evidence of “no effect” in such a small sample.

Thank you for this comment – yes – we corrected it (line 466)

Do not report p=0.000 in supplementary table S4 (twice), but rather p<0.001 (as the p-value can never be exactly 0).

It was corrected

Round 2

Reviewer 3 Report

The authors have modified the manuscript according to most of the suggestions. However, there are still some issues left.

Discussion: the discussion should at least clearly state first that there were no (or hardly any) relevant differences in the group comparisons (om3 vs. om6), before elaborating on the pre-post comparisons (week 0 vs week 6 or 12) within the separate groups.

Multiple testing is still not critically discussed, at least not in lines 367-369 as the authors state in their reply. This should be clearly discussed, in particular since the authors even refer to p-values of 0.08 as “marginally significant difference” (lines 387 – 389).

Supplementary table S4 still contains twice a p=0.000, which should rather be p<0.001 (as the p-value can never be exactly 0), even though the authors state that they have corrected this.

The expression “not different” instead of “not significantly different” is still used in lines 551 and 558. Please change, as of course individual values as well as the point estimates for population means are different, but this absence of a significant difference should not be interpreted as evidence of “no effect” in such a small sample.

Author Response

Reviewer No 3:
Comments and Suggestions for Authors/Answers
The authors have modified the manuscript according to most of the suggestions. However, there are
still some issues left.
We thank the reviewer for accepting the majority of the changes and comments made thorough the revised manuscript.
Discussion: the discussion should at least clearly state first that there were no (or hardly any) relevant
differences in the group comparisons (om3 vs. om6), before elaborating on the pre-post comparisons
(week 0 vs week 6 or 12) within the separate groups.
Multiple testing is still not critically discussed, at least not in lines 367-369 as the authors state in their
reply. This should be clearly discussed, in particular since the authors even refer to p-values of 0.08 as
“marginally significant difference” (lines 387 – 389).
We emphasized in the Results section that there were no significant differences in basal values in the
monitored subfractions (line 344 of the modified manuscript with marked changes).
We have added the following text to the Discussion, in which we discuss the effect of
supplementation on LDL (lines 467-471) and HDL subfractions (lines 474-491), we hope, in
accordance with the reviewer's recommendation.
Lines 467-471:
At the baseline, LDL-CH did not differ between the patients and controls. Even though non-atherogenic LDL1
subfraction increased after both, omega-3 and omega-6 FA supplementations, none of the LDL subfractions
showed any correlation with CDI scores at baseline or after supplementation with omega-3 and omega-6 FAs.
Since neither LDL-CH nor LDL subfractions correlated with CDI scores, the predicted association with depressive
disorder in children and adolescents seem debatable.
Lines 474-491:
In our study, omega-3 FA supplementation significantly affected the levels of more HDL subfractions, in contrast
to LDL subfractions, after 6 and 12 weeks of supplementation.
Since the basal values of individual subfractions in the Om3 and Om6 groups were similar, the effects of omega-
3 and omega-6 FA supplementation on HDL subfractions were evaluated using pre-post comparison (week 0 vs.
weeks 6 and 12) in both groups as well by statistical analysis of a significant difference between the Om3 and
Om6 groups in individual HDL subfractions. Significant differences were found for large HDL subfractions, L-HDL2
and L-HDL1+2+3. The marginally significant differences between Om3 and Om6 in L-HDL1 and L-HDL3 should be
explained by the inter-variability and the small number of included individuals in the given groups.
To our knowledge, the relationships between depression severity and lipoprotein subfractions have not been
studied to date. Therefore, it is not possible to confront our results with the results of other authors. In our
previous work, in which hypercholesterolemic children were supplemented with an emulsion of fish oil enriched
with omega-3 FA together with plant sterols and vitamins B6 and B12 for 16 weeks we found a shift in the
direction of the increase of non-atherogenic L-HDL subfractions and decrease of atherogenic HDL subfractions
[23]. Similarly, in our current work, supplementation with omega-3 FAs, but not omega-6 FAs, shifted the
spectrum of HDL subfractions in depressed children in favor of large HDL subfractions (L-HDL1 to 3) after 6 and
12 weeks of supplementation and borderline significant reduced level of small HDL subfraction (S-HDL8) after 12
weeks of supplementation.

Supplementary table S4 still contains twice a p=0.000, which should rather be p<0.001 (as the p-value
can never be exactly 0), even though the authors state that they have corrected this.
We apologize for this. The change in p <0 ...... was made in the revised version, but there was probably a
technical error entering the system. We hope it's okay now.
The expression “not different” instead of “not significantly different” is still used in lines 551 and 558.
Please change, as of course individual values as well as the point estimates for population means, are
different, but this absence of a significant difference should not be interpreted as evidence of “no
effect” in such a small sample.
It was applied in all cases, where justified (lines 433, 454, 467, 492, 504). We're sorry, but in this
the latest version, with the changes highlighted, it's hard to give the line number that the reviewer had in
mind)

This manuscript is a resubmission of an earlier submission. The following is a list of the peer review reports and author responses from that submission.

Round 1

Reviewer 1 Report

Authors

The paper by Katrencikova et al. investigates the relationships between depressive disorder, lipoprotein profile and omega-3 fatty acids. In particular, the data presented show that treatment with omega-3 FAs associates with potentially beneficious modifications in HDL profile.

The topic is interesting and stimulating; however, the patient population is relatively limited and this may affect the overall impact of the paper.

General and major observations:

Even if the Introduction is rather lengthy, the rationale of the study is not well defined and, in my opinion, should be better supported by the literature.

Page 2, line 62: what do the Authors mean, stating that cholesterol is “poorly absorbed”? Indeed, this does not seem to be the case.

Results (page 7): The Authors state that Omega-3 FAs increase ALT (at 6 wk), total and LDL-CH (at 12 wk). On the other hand, HDL-CH increases with omega-6 FAs. How do the Authors explain these findings and their significance?  

Page 7 line 292: the increase in IDL-2 with omega-6 is not significant, looking at Figure 3. Furthermore, this sentence needs to be rewritten (“…also increased…was observed…”).

Page 8, Figures 2 and 3: the effects of omega-3 and omega-6 FAs on LDL profile seem superimposable.

Overall, the presentation of the data appears redundant: in my opinion, the findings in Table 3 might be condensed (e.g., moving PON data to Supplemental Material).

Discussion, page 14, lines 415-433: the extended comments on the inconsistencies of the literature are poorly related to the original data presented in this paper; this part appears largely speculative and rather lengthy.

The part of the Discussion at page 15 (lines 465-475) in my opinion also needs to be shortened.

Minor points and style:

Abstract page 1: line 27: PON should be spelled out.

Same page, line 35: “in the contrary to…” should be modified (e.g., “but not…”).

Page 2, line 75: in the sentence “…meta-analysis of (21)...” probably the name of the Author(s) is missing.

Page 2, last paragraph: the term “lipoproteins” after LDL and HDL sounds redundant

Page 3 line 126: the term “children patients” does not seem appropriate.

Page 9, line 314 and following: the subfractions of large HDL are three, whereas the percent changes described are four (22, 16, 19, 7 %). The findings must be consistent.

Page 13, line 390: change “lipid parameter” to “lipid parameters”

Page 14 line 491: “diaries” is misspelled

Reviewer 2 Report

In their submitted paper, Barbora and colleagues performed a post-hoc analysis from the Depoxin project and assessed the effects of administration of omega-3 and omega-6 fatty acids on lipoprotein subfractions and erythrocyte membrane fluidity in children and adolescents with depressive disorder. In their randomized study, the authors included 60 depressed patients aged 7 to 18 years with normal eating habits and no indication of chronic somatic disease as well as 20 untreated controls. Patients were randomized to receive either an omega-3 fatty acids-rich fish oil emulsion (Om3 group) or an omega-6 fatty acids-rich sunflower oil emulsion (Om6 group) for 12 weeks. Biochemical parameters measured included the lipid profile, enzymatic paraoxonase, and erythrocyte membrane fluidity. The authors reported that, after supplementation of omega-3 fatty acids in depressed patients, subfractions with large HDL increased, whereas small HDL subfractions decreased. From these data, they concluded that HDL cholesterol and its subfractions, in the contrary to LDL, may play a role in the pathophysiology of depressive disorder.

Although the manuscript is well written, the conclusions drawn by the authors are unfortunately not supported by their data. In order to show an effect of Om3 treatment on the lipid profile and membrane fluidity, the authors should have performed a direct comparison between the Om3 and Om6 groups. Since the randomization into the two arms was a key feature of the study design, this important comparison is required. Instead, the authors tested for time-dependent changes within either of the two group, but no data are presented for a between-group comparison. Moreover, in the Supplemental Table S3 the authors showed an increase in LDL-1 and LDL-2 subfractions from baseline to the six-week follow-up for each group. Thus, it is unclear why the authors claimed that Om3 has an effect on HDL but not on LDL. Despite the randomization procedure it may be possible that the two groups differed already at baseline, however this was not investigated, although the data should be available.

In all figures, the histograms presented did not include a standard deviation for each group and time point.

Please do not use unexplained abbreviations in the Abstract such as PON1.

Reference 9 and reference 22 are identical.

The Abstract did not mention the number of study participants.

Supplemental Figure 3 gives a number of 21 control subjects, while 20 should be correct.

----------------